# Influence of Fiber Volume Content on Thermal Conductivity in Transverse and Fiber Direction of Carbon Fiber-Reinforced Epoxy Laminates

**DOI:** 10.3390/ma12071084

**Published:** 2019-04-02

**Authors:** Simon Bard, Florian Schönl, Martin Demleitner, Volker Altstädt

**Affiliations:** Department of Polymer Engineering, University of Bayreuth, Universitätsstr. 30, 95444 Bayreuth, Germany; florian.schoenl@uni-bayreuth.de (F.S.); martin.demleitner@uni-bayreuth.de (M.D.); altstaedt@uni-bayreuth.de (V.A.)

**Keywords:** thermal conductivity, prepreg, carbon fiber

## Abstract

Thermal conductivity is an important material property for thermo-mechanical calculations, as mechanical properties strongly depend on the temperature and heat distribution in the manufactured parts. Although several suggestions for approximation formulae have been made, existing experimental data are rare and are not comparable due to different measurement methods. In addition, scarcely has the thermal conductivity in both the fiber direction and transverse direction been studied. The aim of the current research is to show the influence of carbon fiber volume content on the thermal conductivity of laminates. The values are then used to verify the micromechanical models used in the literature. A strong influence on the thermal conductivity could be determined. For the transverse thermal conductivity, the correlation was exponential; for the conductivity in the fiber direction, a linear correlation was found.

## 1. Introduction

The thermal load of the material strongly affects the mechanical properties of fiber-reinforced composites. The distribution of heat in a manufactured part leads to inner stresses and microcracks. An important material constant is, therefore, the thermal conductivity. With that known, approximation formulae for simulations can be used to calculate the heat distribution [1].

Only a few researchers have focused so far on the thermal conductivity of carbon fiber-reinforced laminates. As the properties of fiber-reinforced composites are highly anisotropic, thermal conductivity perpendicular to the fibers (transverse) and in the fiber direction needs to be distinguished [2]. Most publications have focused on the thermal conductivity in one direction rather than both. The only publication mentioning both used different methods to measure the thermal conductivity in the different directions due to the inapplicability of the used methods for both directions [2]. Therefore, laser flash analysis, which is very versatile, was used in the current research. It is also known as the standard method for the determination of thermal conductivity of polymers [1,2,3,4,5,6,7,8,9].

Rolfes and Hammerschmidt determined the transverse thermal conductivity with different fiber morphologies, i.e., Polyacrylonitrile (PAN) and pitch-based carbon fibers, but the experimental data measured by steady-state guarded hot plate (GHP) only showed transverse thermal conductivities for two different fiber volume content [1]. Pilling et al. prepared laminates with four different fiber volume contents of PAN-based fiber and measured via transient hot-strip (THS) [3]. The results from Rolfes and Pilling differ significantly, which can be attributed to the different measurement methods used in their work. Very interesting results could also be found by Shim et al., who showed the influence of the fiber shape on the thermal conductivity for pitch-based fibers [4]. The transverse thermal conductivities from the literature are plotted in Figure 1.

Only a few publications that deal with thermal conductivity in the fiber direction could be identified. Yu et al. measured a thermal conductivity of 1.8 W/mK in fiber direction with a fiber volume content of 48.5 vol% [5]. Evans et al. found a value of 7.2 W/mK with a fiber volume content of 64.6 vol% [6]. In both publications, PAN-based fibers were used.

The difficulty in the application of formulae for the calculation of transverse thermal conductivity lies in the determination of the transverse thermal conductivity of the carbon fiber. Huang et al. developed a setup in a vacuum chamber of a scanning electron microscope based on stationary heat flow. For a pitch-based fiber (YSH-60, Nippon Graphite Fiber Corporation, Himeji, Japan), they measured a transverse thermal conductivity of 12 W/mK [7]. To our best knowledge, no measurement has so far been conducted on the conductivity of PAN-based carbon fibers. As the morphology and ability of pitch-based fibers and PAN-based carbon fibers to conduct heat differ significantly, the PAN-based fibers are expected to exhibit significantly lower transverse thermal conductivities [8].

## 2. Production and Characterization Methods

### 2.1. Materials

The resin tetraglycidylmethylenedianiline (TGMDA, Epikote^TM^ RESIN 496, Hexion Inc., Columbus, OH, USA) is tetra-functional with an epoxy equivalent of 115 g/eq and was cured with diethyltoluenediamine (XB3473^TM^, DETDA, hydrogen equivalent weight 43 g/eq). A PAN-based fiber HTS40 (Teijin Carbon Europe GmbH, Wuppertal, Germany) with a tensile strength of around 4620 MPa, elongation at break 1.8%, and a Young’s modulus of 239 GPa was used for the prepreg production.

### 2.2. Resin Preparation and Curing

Resin and hardener were stirred in a stoichiometric ratio of 72:28. The mixture was degassed at 10–20 mbar. The samples were cured under pressure in a laboratory press at 120, 160, and 200 °C, for 1 h at each temperature with a heating rate of 10 K/min. A post-curing step at 220 °C for 2 h followed before cooling at a rate of 5 K/min.

### 2.3. Prepreg Production

The unidirectional prepregs were produced via hot-melt processing at the laboratory-scale prepreg machinery at the University of Bayreuth.

First of all, the unidirectional rovings of 12 K carbon fibers are pre-spread, as shown in Figure 2. Resin film was coated at 25 °C on siliconized carrier paper in the coating unit of the prepreg machinery. Finally, resin film and pre-spread fibers were impregnated to the final prepreg with a calander (25 °C). The prepregs were then further processed via hand-layup to the final unidirectional prepreg laminates and cured with the same parameters as the neat and filled resin samples. The laminate thickness with 18 prepreg layers was 2 ± 0.1 mm.

### 2.4. Morphological Characterization

All samples were scanned with a Skyscan 1072 micro-CT (Bruker, Artselaar, Belgium) with a linear resolution of 3.50 µm at a magnification of 80 with an accelerating voltage of 80 kV and tube current of 122 µA. Projection images were acquired over 180° at angular increments of 0.23° with an exposure time of 2.57 seconds per frame, averaged over six frames. Three-dimensional images were reconstructed using the reconstruction software provided by the manufacturer (NRecon Version 1.6.4.1, Micro Photonics Inc., Allentown, PA, USA), where the ring artifact reduction was applied as needed.

Samples were sputtered using a Cressington 108 auto sputter coater with Au coating thickness of 13 nm and studied in a JEOL JSM 6510 scanning electron microscope (JEOL, Tokyo, Japan).

### 2.5. Thermal Conductivity Measurements

The thermal conductivity was measured by the laser flash analysis (LFA) with LFA447 (Netzsch Gerätebau GmbH, Selb, Germany). Five shots were used with a duration of 30 ms each; the signal was fitted with the Proteus Analysis Software (Netzsch Gerätebau GmbH, Selb, Germany) by the Cape–Lehman algorithm. The tested samples had a diameter of 12.7 mm. The laser flash analysis is the standard instrument for the determination of thermal transport properties of carbon fiber-reinforced polymers because of its convenience, short experiment times, and large measurement range [1,9]. For the measurement of fiber direction, the laminates were cut, turned 90°, glued together, and the ends polished to achieve a plane surface. The density was measured with an AG245 (Mettler-Toledo International Inc., Columbus, OH, USA) by Archimedes’ principle. Thermal heat capacity was measured by DSC 1 (Mettler-Toledo International Inc., USA) according to ASTM E1269–11 [10] with a heating rate of 20 K/min.

### 2.6. Determination of Fiber Volume Content

Thermogravimetric measurements were conducted with a TG 209 F1 Libra (Netzsch-Gerätebau GmbH, Selb, Germany).

For the determination of fiber volume content, a routine suggested by Monkiewitsch was followed, which was verified in a very thorough investigation [11]. Additionally, the applicability was verified in pre-experiments and compared to the results from ashing the samples for 24 h at 600 °C in a muffle oven and a second measurement by wet chemical methods following DIN EN 2559. In the suggested routine, fibers were dried for 2 h at 120 °C in the thermogravimetric analysis (TGA). Then, the fibers were heated in TGA from 20 °C to 800 °C with a heating rate of 2 K/min under nitrogen flux of 85 mL/min. The samples of the laminates and samples from the neat resin were also dried for 2 h in TGA, and then heated to 450 °C with a heating ramp of 10 K/min. Finally, an isothermal step for 170 min at 450 °C followed. All samples were handled with gloves to prevent possible contamination.

Additionally, the density was measured with an AG245 (Mettler-Toledo International Inc., Columbus, OH, USA) by Archimedes’ principle. In addition, by density measurements, the fiber volume content could be determined, but this method delivers unreliable results as the porosity significantly influences the measurement. 

## 3. Results and Discussion

### 3.1. Fiber Volume Content and Morphology

In the TGA, the fibers showed only a slight weight loss of 1.0 ± 0.1%, which can be attributed to the oxidation of the sizing. During the drying step, no significant weight loss could be detected. 

The fiber volume content ϕ could then be calculated to:(1)$$\phi =\frac{m_{f}}{\rho_{f}\left(\frac{m_{f}}{\rho_{f}}+\frac{1-m_{r}}{\rho_{r}}\right)}$$
where $m_{f}$ is the mass of the fibers, $\rho_{f}$ is their density, and $\rho_{r}$ represents the density of the resin. The mass of the fibers was calculated by
(2)$$m_{f}=\frac{m_{l}-m_{r}}{1-m_{r}}$$
where $m_{l}$ is the remaining mass of the laminate after the cycle, and $m_{r}$ is the remaining mass of the resin. Figure 3 shows the weight loss of the resin and laminate during TGA. From the resin sample, the remaining relative weight $m_{r}$ was at 17.55 ± 0.10%. The calculated fiber volume contents can be found in the legend of the graph. Standard deviations were neglected as they were < 1% for the three measured samples.

In the next step, an estimation of the porosity of the laminates was conducted. Therefore, the density of the laminate was determined by Archimedes’ principle and compared to the density of a theoretical non-porous sample with the fiber-volume contents determined in the TGA. The difference between the actual density of the produced laminate and the theoretical density of a non-porous sample is an indicator of the porosity of the samples. As shown in Figure 4, both values do not differ significantly. For the sample with a fiber volume content of 54.9%, the porosity was 1.6%; for all other samples, the porosity was calculated to < 1%.

### 3.2. Heat Capacity, Diffusivity, and Thermal Conductivity

The thermal conductivity can be calculated from the heat capacity $c_{p}$, density ρ, and thermal diffusivity a by the following equation:(3)$$\lambda =\rho *c_{p}*a$$

The densities at room temperature are shown in Figure 4. The heat capacity was determined to 1.11 ± 0.04 J/gK for the resin hardener system and 0.97 ± 0.05 J/gK for the PAN fiber at room temperature. The heat capacity of the resin is in accordance with the findings of Baller [12]. Rana et al. determined the heat capacity of the carbon fiber to 0.92 J/gK, so the measured value seems valid [13]. Knowing these values, the heat capacity of the samples can be calculated. The heat capacity, density, and diffusivity measured by the laser flash method can be found in Table 1.

Figure 5 shows the transverse thermal conductivities of the samples determined by LFA. It should be mentioned that the standard deviations are not visible as they are smaller than 0.015 W/mK. Additionally, the thermal conductivities taken from the publications of Pilling et al. [3] and Rolfes and Hammerschmidt [1] were normalized by the thermal conductivity of the neat resin. The scaling factor is the thermal conductivity of the resin in the publication divided by the thermal conductivity measured in the current research. The value at 0 vol% of fiber is the thermal conductivity of the neat resin. The data from the current research is, therefore, in good agreement with the data presented in the mentioned publications.

The presented normalized thermal conductivities showed minor deviations. It can be concluded that the transverse thermal conductivities of the carbon fibers used in the publications and the current research are very similar.

Figure 6 shows the axial thermal conductivities of the laminates with varying fiber volume content. The thermal conductivities in the fiber direction are expected to follow the rule of mixture as no interfaces between the fibers and matrix can be expected to influence the thermal flow. With the volume contents of the components $\phi_{i}$ and their thermal conductivity $\lambda_{i}$, the thermal conductivity of the compound λ is calculated as:(4)$$\lambda =\underset{i}{\sum }\phi_{i}*\lambda_{i}$$

A linear fit following the equation f (x) = 0.23 + 0.0544x was used to demonstrate the linear coherence. The R^2^ is at 0.9991, which shows an excellent correlation of the formula. A linear fit following the equation f (x) = 0.23 + 0.0544x was used to demonstrate the linear coherence. With the formula above, a thermal conductivity of the carbon fiber in the fiber direction of 5.63 W/mK can be calculated. The manufacturer of the specific fiber used in the underlying research does not provide data for the thermal conductivity, but Hexcel mentions thermal conductivities for fibers IM7, IM10, and AS4 of 5.4, 6.1, and 6.8 W/mK [14].

## 4. Micromechanical Calculations

In the following section, the experimental data was used to test several models to predict the thermal conductivity of the composites. A model based on calculations by Rayleigh [15] and a self-consistent formula presented in the work of Rolfes and Hammerschmidt [1] were compared. Then, a model by Lewis and Nielsen [16], which is widely used in composites of polymers and filler, was applied.

The formula of the self-consistent model is:(5)$$\lambda_{c}=\frac{\lambda_{f}+\lambda_{m}+\left(\lambda_{f}-\lambda_{m}\right)\phi }{\lambda_{f}+\lambda_{m}-\left(\lambda_{f}-\lambda_{m}\right)\phi }\lambda_{m}$$
while ϕ is defined as the filler content, $\lambda_{f}$ as the conductivity of the fiber, and $\lambda_{M}$ as the conductivity of the matrix.

The equation of Rayleigh is:(6)$$\lambda_{c}=\lambda_{m}\left(1-2\frac{\phi }{v^{\prime }+\phi -\frac{3\phi^{4}}{v^{\prime }\pi^{4}}S_{4}^{2}}\right)$$
with
(7)$$v^{\prime }=1+\frac{\frac{\lambda_{f}}{\lambda_{m}}}{1-\frac{\lambda_{f}}{\lambda_{m}}}$$
and $S_{4}=0.0323502\pi^{4}$ as suggested by Rolfes et al. [1].

The formula of Lewis and Nielsen was used in various publications to calculate the thermal conductivity of the composites [1,9,17,18]. It is formulated as follows:(8)$$\lambda_{k}=\lambda_{M}\frac{1+AB\phi }{1-B\phi C}$$
while ϕ is defined as the filler content and $\lambda_{M}$ as the conductivity of the matrix. The factors A,B and C reflect the filler geometry, orientation and intrinsic thermal conductivity. According to Guth [19], *A* can be calculated with the aspect ratio p of the filler:(9)$$A=\frac{p}{\left[2\ln\left(2p\right)\right]-3}+1$$

*B* is not an independent variable as it also reflects conductivity of the filler and matrix and *C* reflects the maximal packing density $\phi_{max}$:(10)$$B=\frac{\frac{\left(\lambda_{F}\right)}{\left(\lambda_{M}\right)}-1}{\frac{\left(\lambda_{F}\right)}{\left(\lambda_{M}\right)}+A}$$
(11)$$C=1+\frac{\left(1-\phi_{max}\right)}{\phi_{max}^{2}}\phi$$

The maximum packing density of the fibers in the composite was 82% [20]. *B* was calculated with the thermal conductivity of the matrix given above and the transverse thermal conductivity of the fiber of 2 W/mK. The transverse thermal conductivity of the fiber can only be estimated as no method to measure a single fiber could be identified in the literature. Rolfes and Hammerschmidt [1] calculated a transverse conductivity of 2 W/mK from their experimental data for round-type PAN fiber. *A* was calculated to 0.83 with an aspect ratio of 0.5 and Rolfes and Hammerschmidt confirm that this value has been suggested in the literature [1]. The conductivities in the fiber direction and the transverse thermal conductivity are expected to differ as the fibers are not homogenous. Carbon fibers based on PAN showed a circumferentially orthotropic structure. Only the outer layer effectively transported the transverse thermal conductivity. An explanation is that the thermal resistivity between the different layers is high due to phonon scattering at the interfaces.

Figure 7 shows the experimental and analytical data of the transverse thermal conductivities. The Rayleigh and the self-consistent equation led to quite similar results. The equations suggested by Lewis–Nielsen led to higher values. All equations underestimated the experimental data. For a higher fiber content, the predicted values of Lewis-ielsen were in good agreement with the experimental data.

## 5. Conclusions

The aim of this work was to explore the influence of fiber volume content on the thermal conductivity of carbon fiber-reinforced prepreg laminates.
The transverse thermal conductivity shows non-linear behavior and the formulae presented by Lewis–Nielsen are in excellent agreement with the experimental data. It seems the transverse thermal conductivity of 2 W/mK suggested by Rolfes and Hammerschmidt is also applicable for the carbon fibers investigated in the current research.From the experimental data, it is suggested that the thermal conductivity in the fiber direction follows a linear correlation.

The current research delivers information for the simulation of the thermo-mechanical behavior of carbon-fiber-reinforced composites. It would be interesting to investigate the behavior of fibers with different morphologies, as pitch-based fibers, or metal-coated fibers in future publications. Furthermore, PAN-based carbon fibers might differ in their morphologies and, therefore, show different transverse thermal conductivities. It cannot be concluded from the current research whether the information can also be transferred to different types of carbon fibers.

## Figures and Tables

**Figure 1 materials-12-01084-f001:**
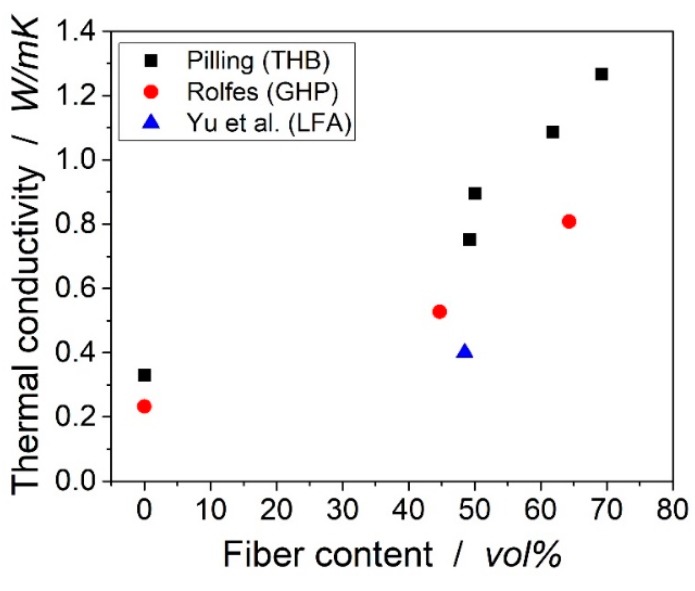
Transverse thermal conductivities taken from the publications mentioned in the text. Measurement methods in brackets. The used matrices are epoxy resins.

**Figure 2 materials-12-01084-f002:**
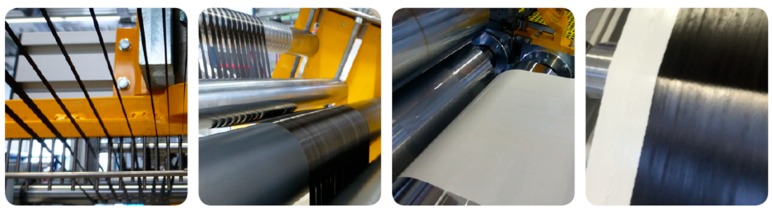
Single undirectional-rovings, pre-fiber spreading unit, resin coating unit, and final prepreg, respectively.

**Figure 3 materials-12-01084-f003:**
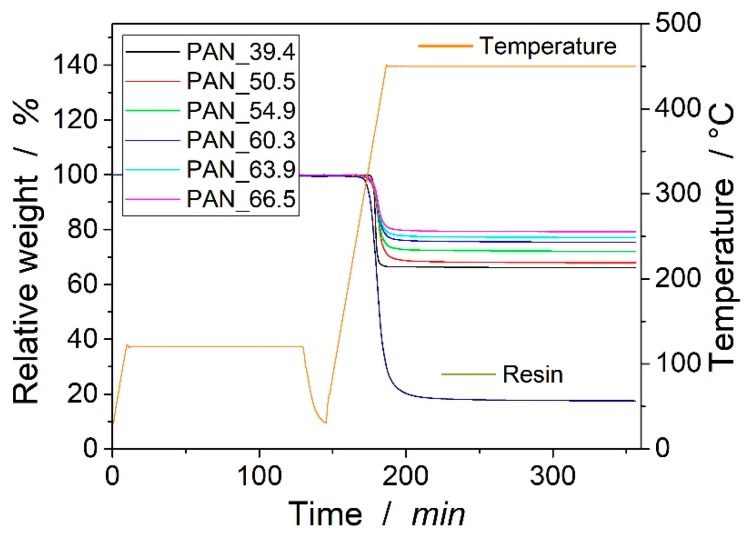
Measurements from thermogravimetric analysis of resin and laminates. Samples are named by the calculated fiber volume content.

**Figure 4 materials-12-01084-f004:**
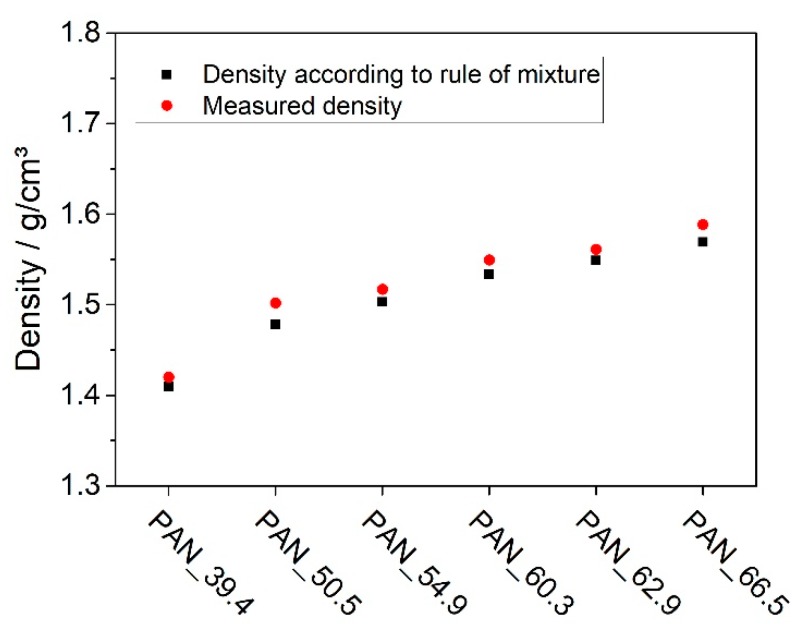
Densities according to the rule of the mixture with fiber volume contents from TGA compared to densities measured and calculated by Archimedes’ principle.

**Figure 5 materials-12-01084-f005:**
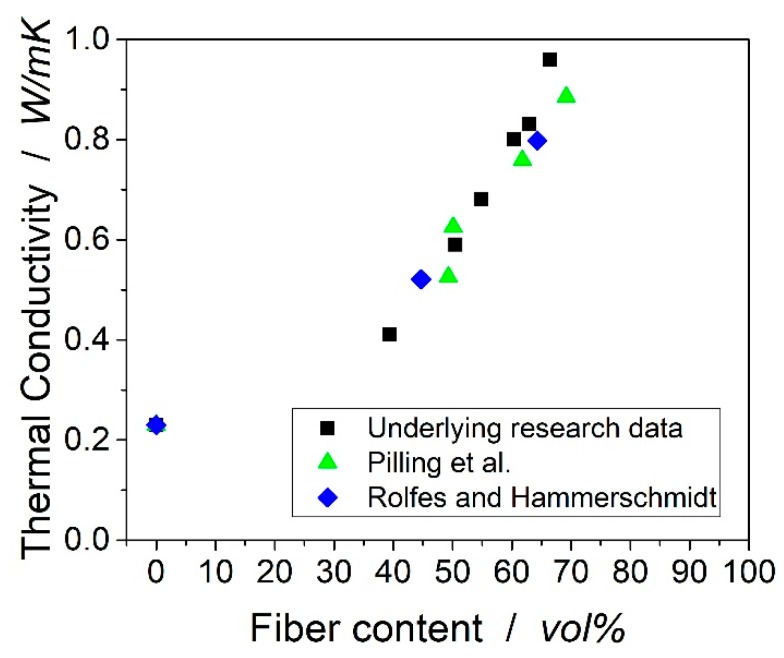
Transverse thermal conductivity of samples measured by the laser flash method compared to experimental data from the literature normalized to the conductivity of the neat resin (0.23 W/mK).

**Figure 6 materials-12-01084-f006:**
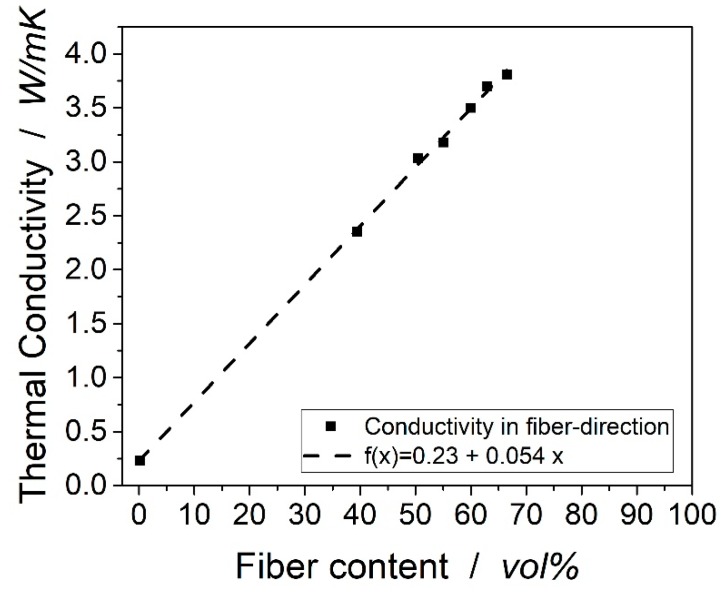
Influence of fiber volume content on thermal conductivity in the fiber direction.

**Figure 7 materials-12-01084-f007:**
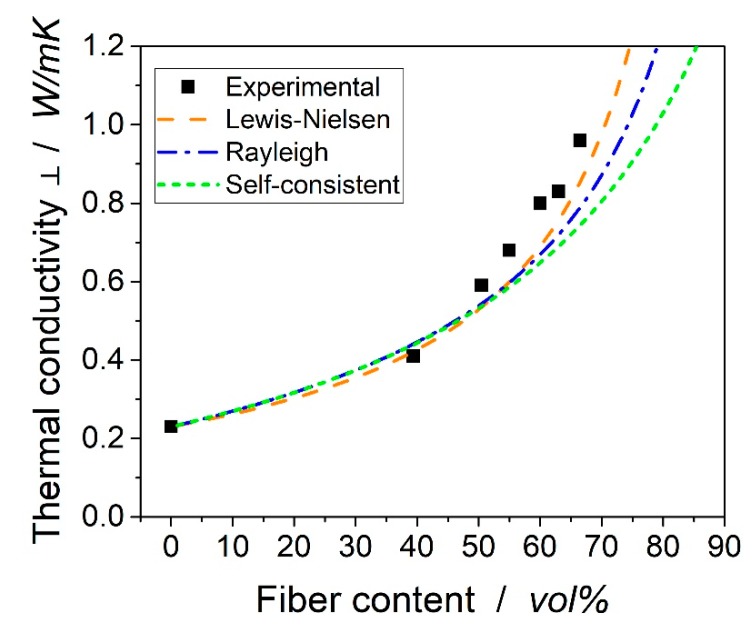
Experimental data versus analytical data from the presented models.

**Table 1 materials-12-01084-t001:** Heat capacity, density, and thermal diffusivity of the tested samples at 20 °C.

Sample	Heat Capacity	Density	Diffusivity ⊥	Diffusivity ||
	J/gK	g/cm³	m²/s	m²/s
Resin	1.440	1.20 ± 0.011	0.135 ± 0.004	
PAN_39.4	1.053	1.39 ± 0.010	0.290 ± 0.005	1.60 ± 0.005
PAN_50.5	1.037	1.45 ± 0.012	0.394 ± 0.003	2.01 ± 0.004
PAN_54.9	1.030	1.47 ± 0.009	0.450 ± 0.004	2.18 ± 0.006
PAN_60.3	1.023	1.50 ± 0.014	0.516 ± 0.007	2.28 ± 0.007
PAN_62.9	1.019	1.51 ± 0.008	0.538 ± 0.005	2.40 ± 0.003
PAN_66.5	1.014	1.53 ± 0.011	0.613 ± 0.008	2.45 ± 0.004

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
