# Peer review of "Influence of Fiber Volume Content on Thermal Conductivity in Transverse and Fiber Direction of Carbon Fiber-Reinforced Epoxy Laminates"

_materials, 2019, doi:10.3390/ma12071084_

Round 1

Reviewer 1 Report

Thank you for the submission. The experimental data will be useful for the literature when you can expand a little more on the sample preparation. 

The English is understandable, but could be improved with some light grammatical editing.

Section 2.3

Please comment on the sample specimens and sample preparation. 

Sample thickness, layers of fiber, etc.

Section 2.5

Please comment on the LFA sample preparation, particularly how you prepared fiber-direction samples

Line 41

What type of carbon fiber was measured by Yu and Evans?

Line 151

"It mentioned that the standard deviations are not visible as they are lower than 0.15 W/mK."

0.15 W/m-K would be visible on Figure 5 - that's bigger than the minor tick marks. Is this correct?

Line 152

  The conductivities were normalized based on the resin. I think this merits more explanation. You corrected the conductivity based on the ratio between their resin and yours?

Line 174:

The self-consistent model you mention is also referred to as the Maxwell-Eucken model (1870). Did you have a reason to call it the "self-consistent model"?

Line 179:

S=0.0323502*pi^4  Is the constant based on geometry? If so, please write out the dimensions you used for the calculation.

Line 165:

I assume the 0.23 then is the measured conductivity of your resin. Can you make that more explicit?

You calculate a thermal conductivity for PAN as 5.63 W/m-K. How does that fit with literature?

Equation (8)

What is p?

Line 188: The maximum packing density is 78%

Can you provide a reference or justification for this?

Maximum circle packing in a plane is about 91%. This is important, because Phi_Max can be used as a fitting factor, instead of a predictive factor.

You can refer to the following reference. Hexagonal close packed fibers -> Phi_MAX=0.907 

Random close packed 2D fibers, Phi_Max = 0.82

Boudenne, L. Ibos, and Y. Candau, “Thermophysical properties of multiphase polymer systems,” in Handbook of Multiphase Polymer Systems, West Sussex, 2011, pp. 403–439.

Section 4. It's great to show the different models for transverse conductivity. Why did you not show them for fiber-direction thermal conductivity? It is linear, but that's not what every model would predict.

Conclusion

Line 214. Fiber-direction can be assumed to be 4 W/m-K

It's not correct to give it a constant value of 4. Maybe you mean ~4 at 67 %vol?

You mention micro-mechanical properties several times. What else do you need to measure to to inform these simulations?

Author Response

Thank you very much for the review. I corrected all the Typos and added additional information where suggested.

The English is understandable, but could be improved with some light grammatical editing.

Section 2.3

Please comment on the sample specimens and sample preparation. 

Sample thickness, layers of fiber, etc.

Section 2.5

Please comment on the LFA sample preparation, particularly how you prepared fiber-direction samples

Line 41

What type of carbon fiber was measured by Yu and Evans?

Line 151

"It mentioned that the standard deviations are not visible as they are lower than 0.15 W/mK."

0.15 W/m-K would be visible on Figure 5 - that's bigger than the minor tick marks. Is this correct?

ð  std is lower than 0.015 W/m-K

Line 152

  The conductivities were normalized based on the resin. I think this merits more explanation. You corrected the conductivity based on the ratio between their resin and yours?

ð  Yes, I made it clear now

Line 174:

The self-consistent model you mention is also referred to as the Maxwell-Eucken model (1870). Did you have a reason to call it the "self-consistent model"?

ð  The term was used by Rolfes and Hammerschmidt

Line 179:

S=0.0323502*pi^4  Is the constant based on geometry? If so, please write out the dimensions you used for the calculation.

ð  The constant is not based on geometry, but comes from Rayleigh formula. The value was suggested by Rolfes at al. and I added a reference for it.

Line 165:

I assume the 0.23 then is the measured conductivity of your resin. Can you make that more explicit?

You calculate a thermal conductivity for PAN as 5.63 W/m-K. How does that fit with literature?

Equation (8)

What is p?

Line 188: The maximum packing density is 78%

Can you provide a reference or justification for this?

Maximum circle packing in a plane is about 91%. This is important, because Phi_Max can be used as a fitting factor, instead of a predictive factor.

You can refer to the following reference. Hexagonal close packed fibers -> Phi_MAX=0.907 

Random close packed 2D fibers, Phi_Max = 0.82

Boudenne, L. Ibos, and Y. Candau, “Thermophysical properties of multiphase polymer systems,” in Handbook of Multiphase Polymer Systems, West Sussex, 2011, pp. 403–439.

ð  I used the value 78% for a cubic packing as it can be seen here (http://www.mse.mtu.edu/~drjohn/my4150/compositesdesign/cd1/cd2.html). But as a random close packed 2D structure might be more appropriate, I changed the value to 0.82.

Section 4. It's great to show the different models for transverse conductivity. Why did you not show them for fiber-direction thermal conductivity? It is linear, but that's not what every model would predict.

ð  It seemed not so interesting to use the models for fiber-direction conductivity as a simple parallel model seemed sufficient enough for the calculation.

Conclusion

Line 214. Fiber-direction can be assumed to be 4 W/m-K

It's not correct to give it a constant value of 4. Maybe you mean ~4 at 67 %vol?

You mention micro-mechanical properties several times. What else do you need to measure to to inform these simulations?

ð  Certainly simulations based on FEM would be interesting, but a elaborative investigation together with graphite-filled matrices will follow here. So this will be addressed in another publication.

Reviewer 2 Report

The manuscript is overall well written and presents new, relevant and interesting data and results. The methodology used in the paper is good, the structure of the paper is intuitive and easy to follow and the data and the results are presented in a proper way. I recommend publication after minor revisions as follows.

1.       In the figure caption of Fig 1 and in the introduction, write explicitly the type of polymer matrix which was used. (Most probably epoxy).

2.       In Table 1, please add the pure polymer diffusivity to the table.  

3.       Line 163. Even though the equation for the rule of mixtures is both simple and well known, it might be good to write it out anyway.

4.       In Figure 5, how is the normalization done? It looks like if the first datapoint of Pilling has been shifted downwards approximately 0.1 W/mK while the last point has been shifted downwards more than 0.2 W/mK. This looks strange!

5.       Line 194. Homogenous or inhomogeneous?

Author Response

1.       In the figure caption of Fig 1 and in the introduction, write explicitly the type of polymer matrix which was used. (Most probably epoxy).

=> Added information as suggested

2.       In Table 1, please add the pure polymer diffusivity to the table. 

=> done

3.       Line 163. Even though the equation for the rule of mixtures is both simple and well known, it might be good to write it out anyway.

=> Added equation

4.       In Figure 5, how is the normalization done? It looks like if the first datapoint of Pilling has been shifted downwards approximately 0.1 W/mK while the last point has been shifted downwards more than 0.2 W/mK. This looks strange!

=> Added information in the text (scaling factor is thermal conductivity of resin in the publication divided by thermal conductivity measured in the underlying research).

5.       Line 194. Homogenous or inhomogeneous?

=> Typo corrected

Reviewer 3 Report

This was a very well rounded paper with concise wording easy to follow throughout. The results supporting past literature and new insight is provided on the back of existing studies. I suggest this paper be accepted after minor revisions as listed below:

Line 7: The thermal conductivity (Remove ‘the’)

Line 10: Experimental data are rare (Change ‘are’ to ‘is’)

Line 20: The temperature strongly influences (The temperature of what? Elaborate)

Line 27 to 29: The few publications (The word publication(s) implies more that one, yet only one reference is provided. Either add more references or reword.)

Line 73: Spacing needed (at_25)

Line 75: Calendar? (Is this correct or a typo? Explain)

Line 87: Samples were sputtered by… (Replace ‘by’ with ‘using a’)

Line 98: Citation to the appropriate ASTM is needed.

Line 137: Figure 4 difficult to compare results as dots are overlaid. Could this be displayed in a more clear way as it is difficult to compare.

Line 165: Could a coefficient of determination (r^2) value be provided on the linear equation to provide completeness.

Line 171: Please reference each author and the exact publication.

Line 181: Should references be before or after the full stop? Either one, please keep consistent throughout.

Line 194: Full stop either side of [1]. Please choose if before or after.

Line 195: Fibers are now homogenous? Is this a typo and meant to say 'not homogenous’?

Line 198: PAN carbon fibers should not be referred to as onion-like as their micro-structure does not represent this at all. PITCH fibers however has been exhibited to be created in onion like structures. To ensure there is no confusion for researchers please alter this sentence and remove the mention of 'onion-like'.

Author Response

Thank you very much for the helpful comments!

I reworked the paper according to the suggestions.

Line 7: The thermal conductivity (Remove ‘the’)

Line 10: Experimental data are rare (Change ‘are’ to ‘is’)

Line 20: The temperature strongly influences (The temperature of what? Elaborate)

Line 27 to 29: The few publications (The word publication(s) implies more that one, yet only one reference is provided. Either add more references or reword.)

Line 73: Spacing needed (at_25)

Line 75: Calendar? (Is this correct or a typo? Explain)

Line 87: Samples were sputtered by… (Replace ‘by’ with ‘using a’)

Line 98: Citation to the appropriate ASTM is needed.

Line 137: Figure 4 difficult to compare results as dots are overlaid. Could this be displayed in a more clear way as it is difficult to compare.

ð  Adapted scaling of the axis

Line 165: Could a coefficient of determination (r^2) value be provided on the linear equation to provide completeness.

Line 171: Please reference each author and the exact publication.

Line 181: Should references be before or after the full stop? Either one, please keep consistent throughout.

Line 194: Full stop either side of [1]. Please choose if before or after.

Line 195: Fibers are now homogenous? Is this a typo and meant to say 'not homogenous’?

Line 198: PAN carbon fibers should not be referred to as onion-like as their micro-structure does not represent this at all. PITCH fibers however has been exhibited to be created in onion like structures. To ensure there is no confusion for researchers please alter this sentence and remove the mention of 'onion-like'.

ð  Thank you very much, I replaced it by circumferentially orthotropic, which is the term Rolfes and Hammerschmidt use.

Reviewer 4 Report

The present paper investigate the effect of fiber volume fraction on the thermal conductivity of carbon fiber reinforced materials. The article is written is well organized manner. I would recommend to accept the article in its present format.

Author Response

Thank you very much for the feedback.

There are no changed recommended.